# Comparative Study on Probabilistic Forecasts of Heavy Rainfall in Mountainous Areas of the Wujiang River Basin in China Based on TIGGE Data

**Haixia Qi [1]** , **Xiefei Zhi [2],\*, Tao Peng [1,2],\*, Yongqing Bai [1] and Chunze Lin [1]**

[1] Hubei Key Laboratory for Heavy Rain Monitoring and Warning Research, Institute of Heavy Rain, China Meteorological Administration, Wuhan 430205, China; qxynl@163.com (H.Q.); 2007byq@163.com (Y.B.); linchunze@whihr.com.cn (C.L.)

[2] Key Laboratory for Aerosol-Cloud-Precipitation of China Meteorological Administration, Nanjing University of Information Science and Technology, Nanjing 210044, China

\* Correspondence: zhi@nuist.edu.cn (X.Z.); pt_mail@sohu.com (T.P.)

**Abstract:** Based on the ensemble precipitation forecast data in the summers of 2014–2018 from the Observing System Research and Predictability Experiment (THORPEX) Interactive Grand Global Ensemble (TIGGE), a comparative study of two multi-model ensemble methods, the Bayesian model average (BMA) and the logistic regression (LR), was conducted. Meanwhile, forecasts of heavy precipitation from the two models over the Wujiang River Basin in China for the summer of 2018 were compared to verify their performances. The training period sensitivity test results show that a training period of 2 years was the best for BMA probability forecast model. Compared with the BMA method, the LR model required more statistical samples and its optimal length of the training period was 5 years. According to the Brier score (BS), for precipitation events exceeding 10 mm with lead times of 1–7 days, the BMA outperformed the LR and the raw ensemble prediction system forecasts (RAW) except for forecasts with a lead time of 1 day. Furthermore, for heavy rainfall events exceeding 25 and 50 mm, the RAW and the BMA performed much the same in terms of prediction. The reliability diagram of the two multi-model ensembles (i.e., BMA and LR) was more reliable than the RAW for heavy and moderate rainfall forecasts, and the BMA model had the best performance. The BMA probabilistic forecast can produce a highly concentrated probability density function (PDF) curve and can also provide deterministic forecasts through analyzing percentile forecast results. With regard to the heavy rainfall forecast in mountainous areas, it is recommended to refer to the forecast with a larger percentile between the 75th and 90th percentiles. Nevertheless, extreme events with low probability forecasts may occur and cannot be ignored.

**Keywords:** TIGGE; logistic regression model; Bayesian model averaging; heavy rainfall in mountainous areas

## 1. Introduction

Due to the nonlinearity and complexity of the atmospheric system, coupled with some unavoidable uncertainties of initial conditions and models, weather forecasting has transformed from a single deterministic method to a multi-value probabilistic method, which not only coincides with the development of meteorological science but also better serves society, and the ensemble forecast is precisely the core technology of this transformation [1,2]. With the continuous development of the ensemble forecast, ensemble forecasting techniques are systematically applied in many national meteorological centers. The issue of predictive uncertainty in ensemble forecasts can be solved through an effective approach of analyzing the probability of a particular event. For some high-impact weather

events, the probability of occurrence is low but the impact is significant once it happens. In order to further break the limitation of predictability and improve the accuracy of weather forecasting, the 14th World Meteorological Congress decided to establish a 10-year international research program in 2003. Currently, there are three database centers of Interactive Grand Global Ensemble (TIGGE) around the world, which collect ensemble forecast products from more than a dozen major forecast centers around the world. One of the aims is to provide the required data for the development of global probability weather forecasts [3–5].

In order to minimize the uncertainty of ensemble forecasting and make full use of forecasting data, in recent years, multi-variate, multi-parameter, and multi-model integrated techniques have been used for the post-processing of probabilistic forecasts. The key idea is similar to Model Output Statistics (MOS) for deterministic forecasts [6–8], which corrects existing forecast, based on past prediction errors of the model. At present, statistical post-processing methods mainly include the probability matched mean [9], the logistic regression method (LR) [10,11], the neighborhood method [12,13], the multi-variate Gaussian ensemble kernel dressing method [14], the object probability prediction method [15], the ensemble pseudo-bias correction [16], the Bayesian model averaging method (BMA) [17–19], and the empirical connection method [20]. The current research results show that the logical regression [21–23], the nonlinear Gaussian fitting method, and the BMA [24–34] can provide comparatively ideal forecast results, and are the main ones studied.

In 2018, the China National Key Research and Development Program Major Natural Disasters Monitoring, Early Warning and Prevention Key Projects Research and Demonstration of Key Technologies for Flood Exceeding the Designed Level in Basins and Comprehensive Responses in Changing Environments required the selection of a mountainous area of the Wujiang River Basin. It aimed to conduct research on heavy precipitation and flood in this area and forecasting techniques for them. Heavy rainfall in the mountainous area are characteristic of sudden events, short-term, high-intensity, local features, and serious disasters, making them even more difficult to accurately forecast. The current model can only work on certain terrain types, therefore, the prediction accuracy of the model in the mountainous area is relatively low [35–37]. Therefore, it is of practical significance to conduct model post-processing research on the precipitation forecast in the mountainous area. With regard to the post-processing technique, if systematic errors of models are constant, only a small number of training samples are needed to correct existing prediction errors. However, systematic errors often vary with different prediction models. Thus, a small number of samples is not suitable. Wilks and Hamill [22] pointed out that the improvement of forecasting techniques caused by different lengths of training periods is more obvious than that caused by different forecasting methods. Therefore, for different basins, lead times, and weather elements, the first step is to conduct a sensitivity test on the length of the model training period. A comparative study was conducted between the BMA and LR forecasting techniques in the Wujiang River Basin. We mainly focused on identifying the advantages of the multi-model post-processing method over the raw ensemble forecast for heavy precipitation events in mountainous areas. Whether the forecasting skill couldbe improved.

This paper is organized as follows. Section 2 introduces the multi-model ensemble BMA and LRpost-processing methods. Section 3 describes the data and verification methods. Section 4 shows a comparative study of the BMA, LR, and raw ensemble prediction system forecasts (RAW) for heavy rainfall prediction in mountainous areas with lead times of 1–7 days.

## 2. Multi-Model Ensemble Post-Processing Methods

### 2.1. The BMA Model

The BMA model used in this paper follows Raftery et al. [17], only briefly described here. It is a statistical post-processing method that generates probability density function (PDF) by combining multiple statistical models for ensemble inference and prediction, which is defined as

$$p(y|f_1,\ldots,f_K) = \sum_{k=1}^{K} w_k g_k[y|(f_k, y^T)] \tag{1}$$

where $g_k[y|(f_k, y^T)]$ is the probability density function (PDF) of the forecast variable $y$ during the training period $y^T$ when the forecast model $f_k$ is under the best forecasting condition and $w_k$ is the posterior probability of each ensemble member being the best forecast during the training period (i.e., weight), and $\sum_{k=1}^{K} w_k = 1$. Precipitation is a discontinuous variable, thus, the Gamma distribution function, which fits the skewed distribution of the precipitation, was adopted for fitting [10]. The research consists of two parts, as follows.

The first part $P(y = 0|f_k)$. computes the probability at zero precipitation as a function of $f_k$ by a logistic regression model. Sloughter et al. [10] believed that the cube root of the forecast $f_k$. was adequate. Thus, $f_k^{1/3}$ was used as the forecasting variable in this study.

$$\text{logit}[P(y = 0|f_k)] \equiv \log \frac{P(y = 0|f_k)}{P(y > 0|f_k)} = a_{0k} + a_{1k} f_k^{\frac{1}{3}} + a_{2k} \delta_k \tag{2}$$

where $\delta_k = 0$ is an indicator and is equal to 1 if $f_k = 0$ and equal to 0 otherwise. $P(y = 0|f_k)$ and $P(y > 0|f_k)$ is the conditional probability of non-precipitation and $P(y > 0|f_k)$ is the probability of nonzero precipitation given the forecast $f_k$. The parameters $a_{0k}$ $a_{1k}, a_{2k}$ are undetermined parameters, estimated by the Newton–Raphson iterative method against the training period.

The second part is thePDF, where the amount of precipitation is not zero. The conditional PDF is in the form of a Gamma distribution, which is defined as

$$g_k(y|f_k) = \frac{1}{\beta_k^{\alpha_k} \Gamma(\alpha_k)} y^{\alpha_k - 1} \exp(-y/\beta_k) \tag{3}$$

$$\begin{aligned} \mu_k &= \alpha_k \beta_k = b_{0k} + b_{1k} f_k^{1/3} \\ \sigma_k^2 &= \alpha_k \beta_k^2 = c_0 + c_1 f_k \end{aligned} \tag{4}$$

where $y$ is considered to be the observed precipitation corresponding to the forecast, $\Gamma$ refers to the Gamma function, $\alpha_k, \beta_k$ are the shape and scale parameters, respectively, $\mu_k$ and $\sigma_k$ are the mean and variance of the Gamma PDF for each ensemble model, respectively. The cubic roots of the observed precipitation and ensemble model forecasting precipitation are taken as dependent, and forecast variables and parameters $b_{0k}$ and $b_{1k}$ are estimated by the linear regression.

$w_k, c_0,$ and $c_1$ are estimated by maximizing the log-likelihood function. The likelihood function of the BMA model is as follows.

It is maximized by the expectation–maximization (EM) algorithm, as in Sloughter et al. [10],

$$l(w_1,\ldots,w_k; c_0, c_1) = \sum_t \log p(y_{st}|f_{1st},\ldots,f_{Kst}) \tag{5}$$

where $p(y_{st}|f_{1st},\ldots,f_{Kst})$ is calculated by Equation (1), $y_{st}$ is the precipitation observations during the training period, $f_{kst}$ is the forecasting value of each ensemble model during the training period, and s, t index space and time in the training period [19].

In summary, according to Equation (1), PDFs of all ensemble members are accumulated on the basis of their respective posterior probability or weight. Then, the PDF of BMA multi-model ensemble precipitation is obtained. The specific formula is expressed in Equation (6),

$$p(y|f_1, \ldots, f_K) = \sum_{k=1}^{K} w_k[P(y=0|f_k)I[y=0] + P(y>0|f_k)g_k(y|f_k)I[y>0]] \tag{6}$$

where the relationship between $P(y=0|f_k)$ and $P(y>0|f_k)$ is shown in Equation (2), the specific formula of $g_k(y|f_k)$ is Equation (3), and $I[]$ is the indicator function; if $y=0$, then $I(y=0)$ equals to 1, $I(y>0)$ equals to 0, while if $y>0$, then $I(y=0)$ equals to 0 and $I(y>0)$ equals to 1.

In this paper, the forecasting method of a sliding training period was adopted. In the method, the training period was set as a sliding window and the model took the latest $N$ days before the forecasting day as the training period. The trained parameters were then applied to the next training period. Thus, the forecasting model was established dynamically; in other words, training was conducted for each day and each station in the study area.

*2.2. The LR Model*

In this paper, LR presented by Wilks et al. [22] is as follows:

$$P(y \geq y0) = \frac{\exp\left(\frac{1}{K}\sum_{k=1}^{K} a_{0k} + a_{1k}f_k^{\frac{1}{3}} + a_{2k}\delta_k^{\frac{1}{3}}\right)}{1 + \exp\left(\frac{1}{K}\sum_{k=1}^{K} a_{0k} + a_{1k}f_k^{\frac{1}{3}} + a_{2k}\delta_k^{\frac{1}{3}}\right)} \tag{7}$$

In Equation (7), y is the model prediction result; y0 is the threshold; $f_k$ is the ensemble forecast model; $a_{0k}$, $a_{1k}$ and $a_{2k}$ are parameters to be estimated; and $\delta_k$ refers to the dispersion of single model ensemble forecast (i.e., the standard deviation).

## 3. Data and Methods of Verification and Evaluation

*3.1. The Study Area and Datasets*

The Wujiang River Basin, especially its middle and upper reaches, is a typical mountain river with many tributaries of a pinnate distribution. The total length of the main stream is 1037 km with a natural drop of 2124 m, and the basin area is 87,920 km$^2$, which is southwest–northeast inclined with a large elevation difference between the east and west. Moreover, the mean annual precipitation is 600 mm, and the mean annual discharge is 50.5 billion m$^3$. The precipitation in the flood season from May to September accounts for 80% of the annual precipitation. However, extreme weather events occur frequently due to the complex hydro-meteorology in this area. Therefore, the mountainous Wujiang River Basin was selected as the study area in this paper (Figure 1).

The datasets are from the TIGGE ensemble forecasts for five summers (from June to August) of 2014–2018. The multi-member 24 h precipitation forecast data from four centersECMWF(European Centre for Medium Range Weather Forecasts), UKMO(United Kingdom Met Office), CMA(China Meteorological Administration), and JMA(Japan Meteorological Agency)with lead times of 1–7 days and an initial time of 00:00 UTC were used in this study (Table 1). The data from UKMO model in August 2014 and from CMA model in 10 to 12 July and 5 to 7 August 2017 are missing. Thus, data during these two periods were replaced by NCEP (National Centers for Environmental Prediction)global ensemble forecasting system (GEFS) data. This simple data replacement method is similar to that in the current business operation, which aims to run the program smoothly without a lot of data. The dataset can be downloaded from the web page (https://apps.ecmwf.int/datasets/data/tigge/levtype=sfc/type=pf/).

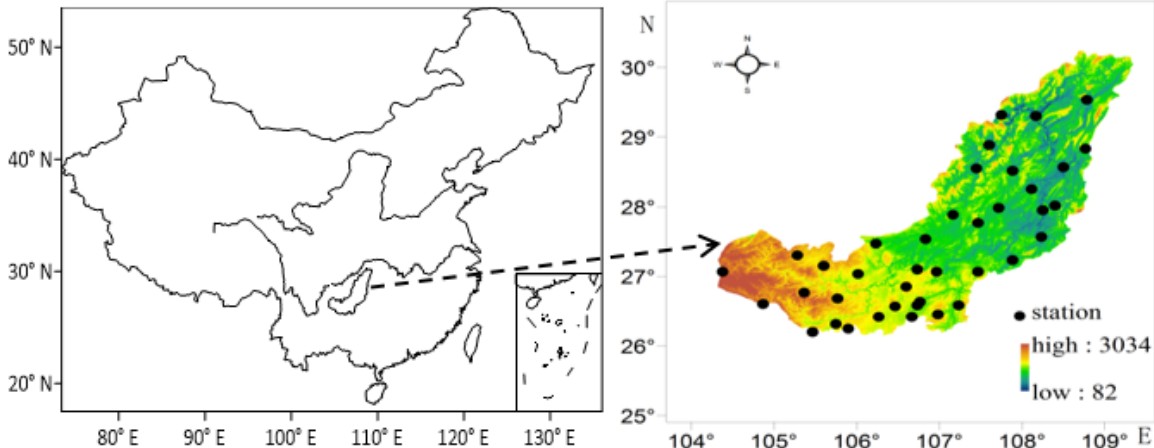

**Figure 1.** Locations of 42 meteorological stations (dots) in the Wujiang River Basin in China.

**Table 1.** Ensemble prediction systems of different centers.

| Centers | Country | Model Spectral Resolution | Ensemble Members (Perturbed) | Spatial Resolution | Forecast Length (Days) |
|---------|---------|---------------------------|------------------------------|--------------------|------------------------|
| ECMWF | Europe | T399L62/T255L62 | 50 | 0.5° × 0.5° | 15 |
| UKMO | United Kingdom | | 11 | 0.5° × 0.5° | 7 + 6 h |
| CMA | China | T213L31 | 14 | 0.5° × 0.5° | 15 |
| JMA | Japan | | 26 | 0.5° × 0.5° | 11 |
| NCEP | America | T126L28 | 20 | 0.5° × 0.5° | 16 |

Based on the ensemble prediction result of each center's model output, first, the result of each multi-member ensemble was averaged, and then the bilinear interpolation method was used to obtain the prediction results of all the stations in two basins. The test period lasted from 21 July to 31 August 2018.

The observation data is the 24 h accumulated precipitation of 42 national meteorological stations in the Wujiang River Basin from 1 June to 7 September during 2014–2018.

### 3.2. Verification and Evaluation Methods

In this paper, the mean absolute error (MAE), Brier score (BS), and the ranked probability score (RPS) were selected to evaluate the ensemble model precipitation forecast. MAE is an indicator reflecting the forecast error. It is calculated as follows:

$$\text{MAE} = \frac{1}{N} \sum_{i=1}^{N} |f_i - o_i| \qquad (8)$$

where $o_i$ is the observed value, $f_i$ is the deterministic forecast result of BMA, which is the median of the BMA predictive PDF, and N is the total number of stations and days.

The formula of BS is as follows:

$$\text{BS} = \frac{1}{N} \sum_{i=1}^{N} (F_i - O_i)^2 \qquad (9)$$

where $F_i$ is the forecasted probability of weather events, $O_i$ indicates the observation, and $O_i$ equals to 1 when the event occurs, otherwise $O_i$ equals to 0, N is the number of forecast times for the two different scenarios.

The RPS was introduced [38] in this research. The RPS is an indicator for a comprehensive evaluation of multiple-level probability prediction results. Suppose a forecast object has a number of J categories, then the RPS scores of each station can be expressed as follows:

$$\text{RPS} = \sum_{k=1}^{J}\left(\sum_{i=1}^{k}F_i - \sum_{i=1}^{k}O_i\right)^2 \tag{10}$$

where $F_i$ and $O_i (i = 1, 2, k...,)$ represent the forecast and observation probability values of different categories k, respectively; $O_i$ equals to either 1 or 0, and only one of these different categories can be 1. The accumulated precipitation Pr is divided into four categories (i.e., J = 4) according to the amount: 0.1 mm ≤ Pr < 10 mm, 10 mm ≤ Pr < 25 mm, 25 mm ≤ Pr < 50 mm and Pr ≥ 50 mm. The RPS value varies in the range of 0–1, and the smaller the better.

The continuous ranked probability score (CRPS) can be regarded as the integration of BS at all possible continuous thresholds. $F_i(x)$, x, and $o_i$ were assumed as the cumulative distribution function (CDF), the precipitation prediction, and the precipitation observation, respectively,

$$\text{CRPS} = \frac{1}{N}\sum_{i=1}^{N}\int_{-\infty}^{+\infty}[F_i(x) - H(x - o_i)]dx \tag{11}$$

where $H(x - o_i)$ is Heaviside function. If x < $o_i$, $H(x - o_i) = 0$, if x ≥ $o_i$, $H(x - o_i) = 1$.

## 4. Results

### 4.1. The Length of Training Period

As the ensemble forecasting skill varies in different regions and seasons, different lengths of training periods were selected to conduct the sensitivity test during summers of 2014–2018 with lead times of 1–7 days. The forecast results of two models in the verification period were evaluated by MAE and RPS, respectively. The mean values of the two verification metrics were calculated by averaging the values of all stations at all days.

Figure 2 shows the comparison of training period lengths of MAE and RPS provided by BMA and LR at the lead times of 1–7 days in the Wujiang River Basin. For the BMA (Figure 2a), the training period of 2 years rendered the best effect of BMA probability forecasts. However, the length of the 5-year training period did not reduce the forecasting error effectively. This indicates that the BMA forecasting model is sensitive to the length of the training period and the forecast quality in the relatively recent period. Moreover, a longer training period is not necessarily better for the forecast skill. Figure 2b shows RPS results from LR for 24 h accumulated precipitation and under thresholds of 1 mm, 10 mm, 25 mm and 50 mm, respectively. It can be seen that with the extension of the training period, forecasting skills are improved, and LR shows the best forecasting effect with the training period of 5 years, which also indicates that the LR forecasting model needs more statistical samples than BMA model.

### 4.2. Comparison, Verification, and Evaluation of Different Models

In this paper, the BS and reliability diagram were used to evaluate the probability forecast effect of multi-model ensemble BMA, LR, and RAW. Based on the studies reviewed in Section 4.1, BMA models adopt the training period of 2 years, while the LR model uses 5 years as the training period length. Figure 3 shows BS score resulted from BMA, LR, and RAW for daily precipitation exceeding thresholds of 10 mm, 25 mm, and 50 mm, respectively, with lead times of 1–7 days. For BMA, the forecast probability value required for the calculation of BS is the proportion of the area under the BMA PDF to the right of the precipitation amount. The RAW forecast probability value is calculated as the ratio of the members whose forecasting precipitation exceeds the threshold to the total members (131 members).

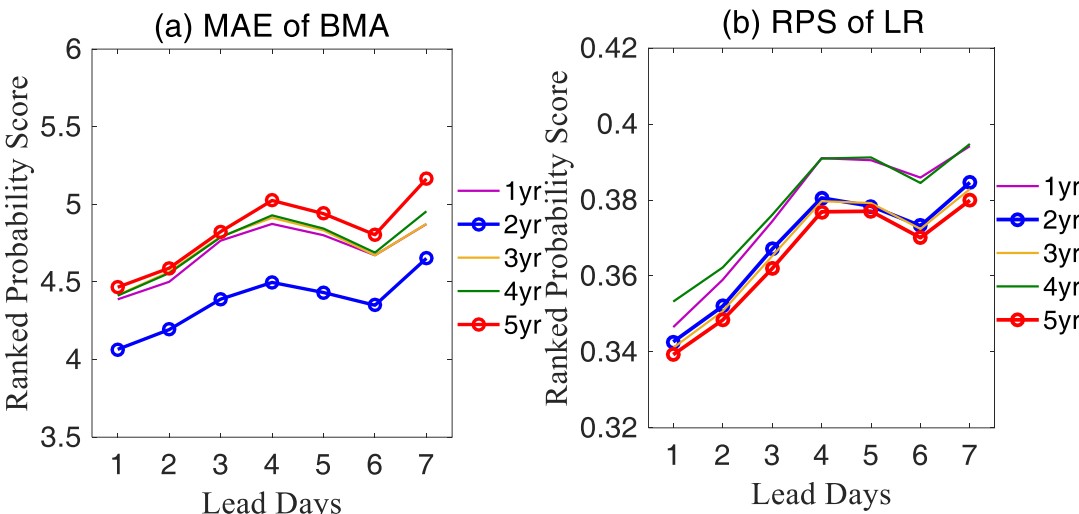

**Figure 2.** Comparison of training period lengths in the Wujiang River Basin: (**a**) The mean absolute error (MAE) of Bayesian model average (BMA) forecasts and (**b**) the ranked probability score (RPS) of the logistic recession (LR) forecasts (1 yr = 1-year training period lengths). Considering the results of the two sensitivity tests, 2 years and 5 years were selected as the optimal training periods for the BMA and LR models, respectively, and then the model parameters were calibrated for the forecast study.

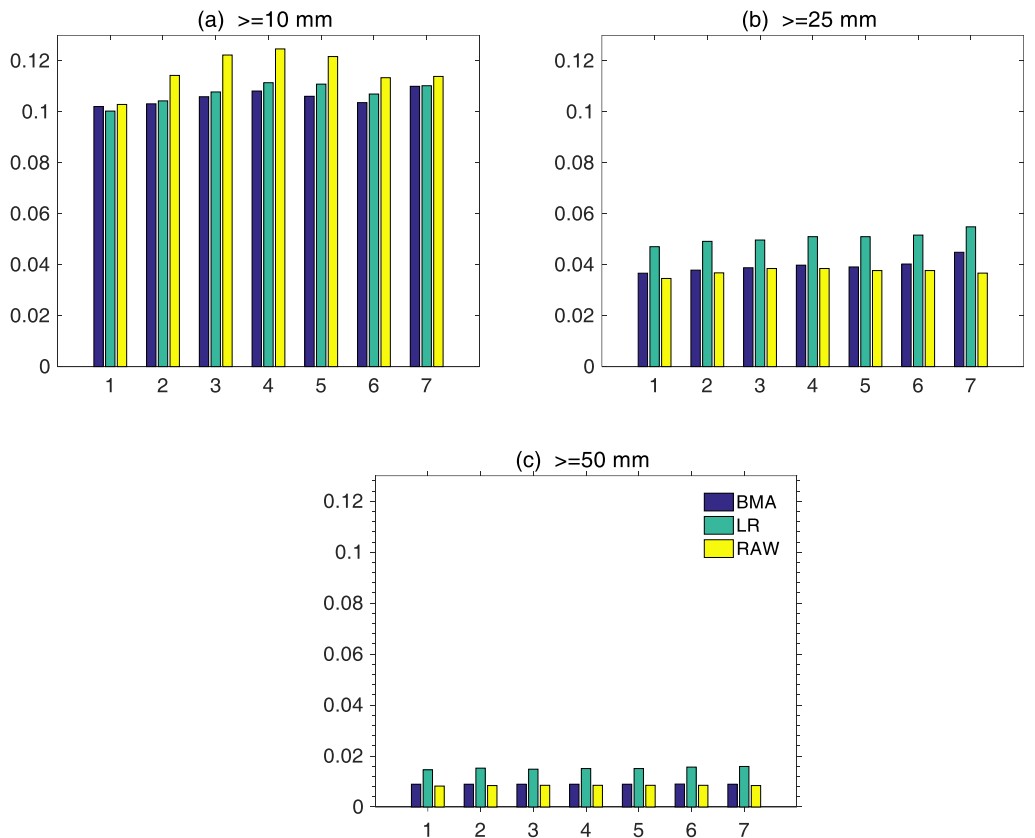

**Figure 3.** The Brier scores of the probabilistic forecasts of 24 h accumulated precipitation exceeding the thresholds of 10 (**a**), 25 (**b**), 50 (**c**) mm with lead times of 1–7 days from Bayesian model average (BMA), logistic recession (LR), and raw ensemble prediction system forecasts (RAW).

As for the precipitation above 10 mm, compared with RAW, the multi-model ensemble methods of BMA and LR have better forecasting effects under the lead times of 1–7 days, while BMA, LR,

and RAW had an equal effect for exceeding the thresholds of 25 mm and 50 mm, indicating that the ensemble method's improvement was not significant. The primary reason for this might be, firstly, the ensemble mean of multi-member ensemble models' results is equivalent to BMA and LR of four models, compared with the large and comprehensive forecasting data of 131 members from the four models, the improvement is considerably limited. Secondly, the BMA method was found to be less effective than the raw ensemble forecast from the lead time of 3 days, indicating that the self-correcting effect of the BMA method was still limited [21].

The reliability diagram refers to the curve of the forecast probability and observation frequency, which more directly reflects the reliability of the probability forecast. The diagonal line in the figure indicates that the forecast probability was equal to the observation frequency, which implies that the probability forecast is completely correct. The area below the diagonal line means over-forecasting and above the line indicates under-forecasting. Figure 4 illustrates the reliability diagram scores for daily precipitation (≥10 mm and ≥25 mm) of BMA, LR, and RAW with lead times of 1 day and 5 days.

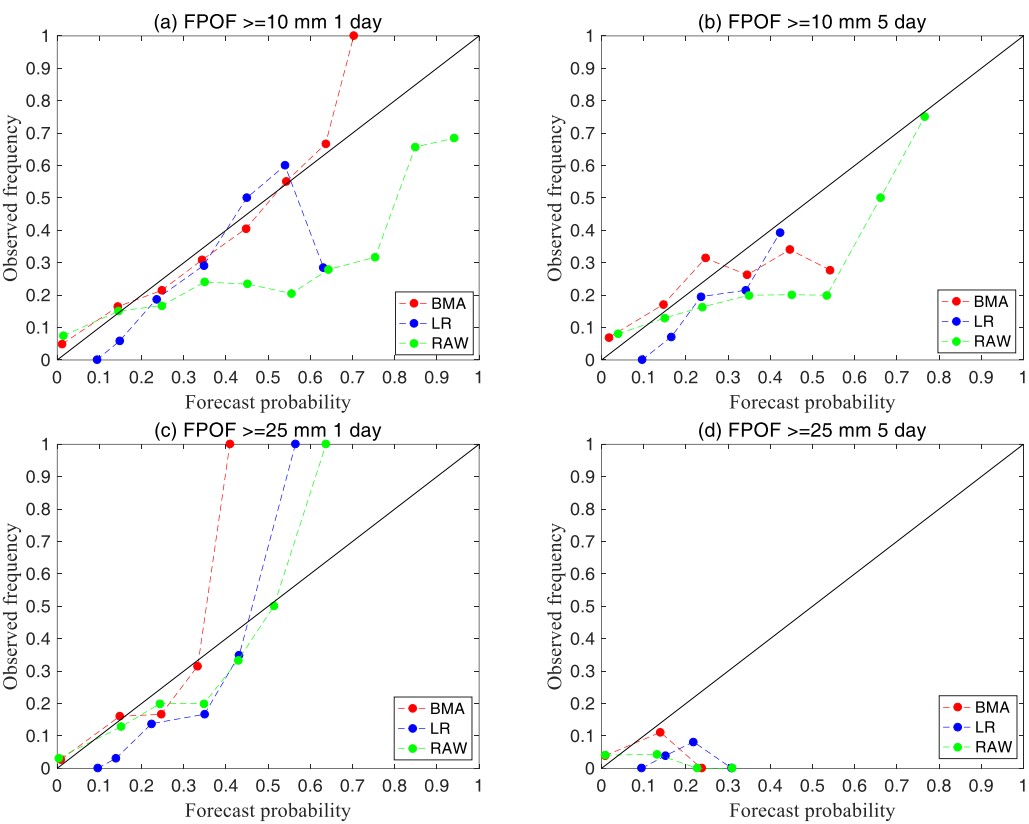

**Figure 4.** The reliability diagram for daily precipitation ≥10 mm (**a**,**b**) and ≥25 mm (**c**,**d**), probability forecasts of BMA, LR, and RAW with lead times of 1 day and 5 days.

All three models show an increasing forecast bias as the lead time and an increase in the precipitation level. The RAW mainly features overestimation of the precipitation probability, especially for the precipitation above 10 mm with a probability between 0.5 and 0.8, which reveals that this forecast method is more likely to overestimate the rainfall above the moderate precipitation level. The LR performed better than the RAW for rainfall exceeding 10 mm, with lead times of 1 day and 5 days (Figure 4a,b), whereas for rainfall events exceeding 25 mm, the LR forecast performance was unstable and was equivalent to the RAW. However, the BMA model shows the best forecasting performance for precipitation of those two levels with the two lead days (Figure 4c,d). It should be noted that for rainfall events exceeding 25 mm with the lead time of 1 day, probability values predicted by BMA and LR are both below 0.5 and no probability value above 0.6 was observed. The results reveal that the uncertainty of the heavy precipitation forecast is often high, and low probability value is likely to

be produced. However, in practical applications, it cannot be ignored that extreme weather events still could happen even when a low probability value is predicted. For forecasting with lead times of 5 days, forecast probability values of the three models are basically below 0.3. This reveals that as the lead time extends, their forecast performances for heavy precipitation worsens, and the prediction ability of the post-processing method is also limited.

Taking the analysis from Figure 4, the forecasts of multi-model BMA and LR were more reliable than the RAW for the heavy and moderate rainfall events. In addition, the BMA model had the best performance in the above four forecasting scenarios. Moreover, as the lead time extended, the model forecast ability for heavy rainfall events decreased, constraining the statistical post-processing of the model output.

*4.3. Case Study of Heavy Rainfall Forecasting*

4.3.1. Comparison of Probability Forecasts

The BMA is a predictive model that creates PDFs and can provide a full probability distribution. In contrast, the LR and RAW can only give the probability forecast at a certain threshold. In this study, a heavy rainfall event, which occurred on 7 August 2018 in the Wujiang River Basin, was selected for forecasting and comparative analyses. Due to the poor forecast for this process by the JMA model, the RAW results were worse than that of the single model. Thus, the single model with the best prediction result was selected for the study.

Figure 5 shows the observed daily precipitation and probability forecasts with precipitation exceeding 25 mm from the BMA, LR, ECMWF, and UKMO with a lead time of 1 day and 5 days (examples of other lead times are omitted) on 7 August 2018. The figure shows that for rainfall events exceeding 25 mm with the lead time of 1 day, BMA model haseffective predictions (Figure 5a–c). On the stations in the southern Wujiang River Basin with the observed rainfall exceeding 25 mm and 50 mm, BMA had an occurring probability over 0.5 and 0.7, respectively (Figure 5a).

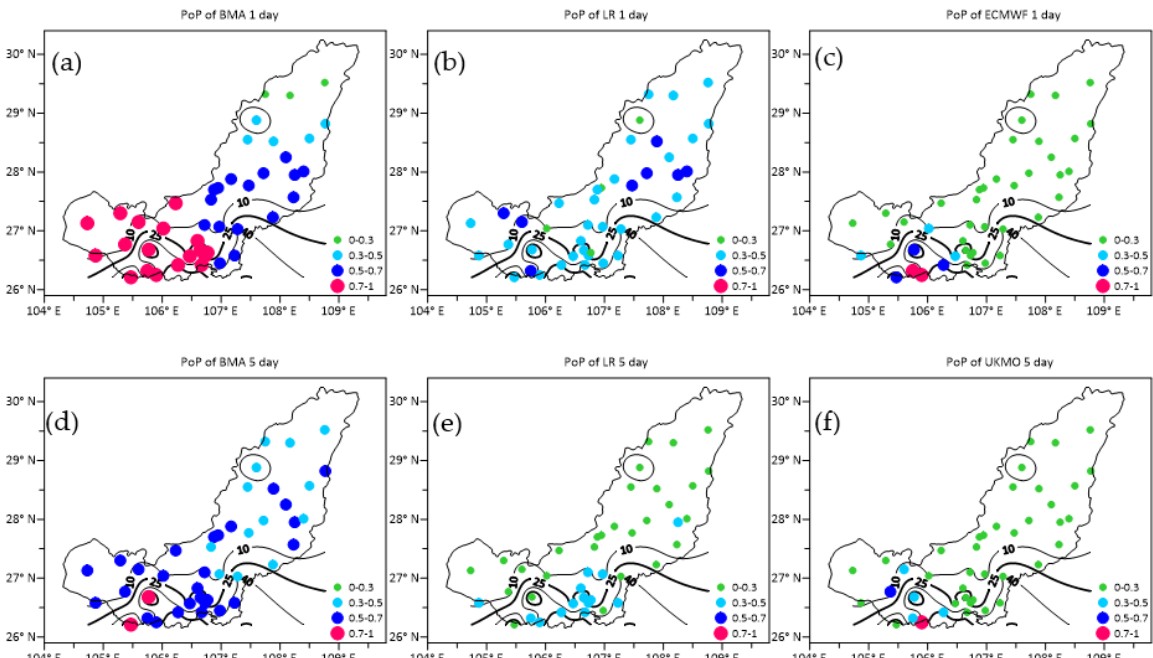

**Figure 5.** The observed daily precipitation and probability forecasts with precipitation exceeding 25 mm from the BMA (**a**,**d**), LR (**b**,**e**), ECMWF (**c**), and UKMO (**f**) with the lead time of 1 day and 5 days on August 7 2018 (the colored dots are forecast probabilities, the black contour is the observed precipitation).

However, the probability predicted by the LR was relatively low, mostly in the range of 0.3–0.5 (Figure 5b). Compared with the BMA model, the area with high probability values above 0.7 predicted by ECMWF was smaller, and low probability values below 0.3 were quite different from the observations. For the lead time of 5 days, the BMA also outperformed the LR and the UKMO (Figure 5d–f). However, for weak precipitation events at some stations in the mid-northern Wujiang River Basin, the BMA model over-estimated the occurring probability of heavy precipitation at both 1-day and 5-day leading times. Meanwhile, the LR and the UKMO forecasted a smaller probability value, which was closer to the observations.

Based on the above analysis, the multi-model BMA method outperformed the LR and RAW, but overestimation of heavy precipitation events still occurred. Forecasts of normal precipitation events from the LR and single model are closer to the actual situation.

The BMA probabilistic forecast can produce a highly concentrated full PDF curve and can also provide deterministic forecasts, whereas the LR and RAW can only produce a probability forecast at a certain threshold. The sharp BMA predictive PDF curve denotes that the forecast actually narrows down all the possible weather events and that the forecast is more reliable. In addition, the BMA also contains quantitative estimates of ensemble forecast uncertainties, which can help analyze extreme events such as heavy precipitation.

For the selected Nayong station, the observed precipitation was normal (1 mm). The BMA deterministic forecast (red diamond) was closer to the observed precipitation (black dot) than the ensemble average forecasting (blue dot). According to the analysis of the CDF curve, the probability of precipitation from BMA exceeding 1 mm was 0.69, the RAW was equal to 1. The probability from BMA and RAW at this threshold were all predicted successfully, but the predicted probability of BMA dropped rapidly to less than 0.5 exceeding the thresholds of 8 mm (2ˆ3 mm) (observed precipitation is 1 mm), while the median probability of RAW was around 0.9. The predicted probability of RAW was higher and the error was larger (Figure 6a). In the case of heavy precipitation events, the observed accumulated precipitation at station 58431 (Liuzhi station) was 60.6 mm, which is higher than the results obtained by ensemble average forecasting and the BMA deterministic forecasts. The forecast results were close to the 90th percentile of BMA predictive PDF, and the probability from BMA and RAW exceeding 60.6 mm was around 0.1. Although the probability was low, it provides certain information about the occurrence of heavy precipitation without any omission. Yet, for heavy rainfall forecast, it is more reasonable to refer to the forecast at 75th–90th percentile of the BMA PDF (Figure 6b).

4.3.2. The Deterministic Forecast

The BMA method can provide the corresponding deterministic forecast by analyzing percentile prediction results of the PDF curve. In the above cases, Figure 7 shows the observed precipitation distribution, ensemble average forecasting, single model, and the 50th percentile and 90th percentile of BMA forecasts. The colored dots represent the levels of 24 h accumulated observed precipitation, including the light rain (0–9.9 mm, green dots), moderate rain (10–24.9 mm, lake blue dots), heavy rain (25–49.9 mm, blue dots) and rainstorm (50–100 mm, pink dots). On 7 August, heavy rains occurred in the upper reaches of the Wujiang River Basin, especially in the western mountainous area. Rainstorm events were observed at four stations (Figure 7a). With regard to the forecast of heavy precipitation centers, the ensemble average forecast (AVE) from the four center models was about 20 mm, which displays a large difference from the observed precipitation (Figure 7b). In general, the overall forecast of the precipitation was comparatively low, and the forecasts from the ECMWF and UKMO were relatively better, as they are sensitive to the events above the threshold of heavy rain (Figure 7c, e). The forecast of the BMA at the 50th percentile was more accurate with light rain, yet the precipitation prediction with heavy precipitation was still lower, and there was little improvement compared with the single-model forecast (Figure 7g). The 90th percentile forecast is generally used as the upper limit of the BMA forecast, which clearly indicates the probable level of heavy precipitation to occur (Figure 7h).

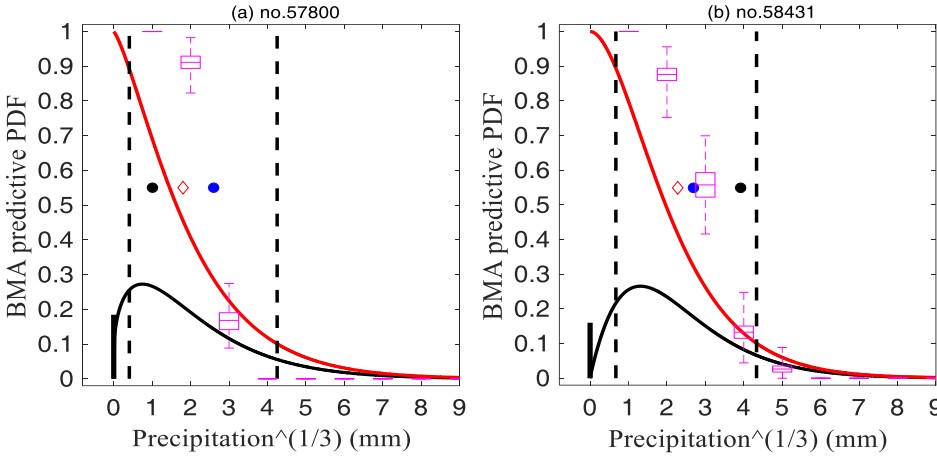

**Figure 6.** The BMA total probability density function (PDF) from BMA and the box plot of RAW probability forecasts statistical distribution calculated by bootstrapping (resampled 1000 times) for 24 h accumulated precipitation at the 24 h lead time at station no. 57800 (**a**) and no. 58431 on 7 August 2018 (**b**). The upper and lower ends of the boxes are the upper and lower quartiles, respectively; the bars through the boxes are the medians; and the upper and lower ends of the whiskers are the maxima and minima, respectively. The thick black vertical line is the discrete probability of non-precipitation. The red curve is the cumulative probability distribution function (CDF). CDF is the proportion of the area under the BMA PDF to the right of the precipitation amount, multiplied by the probability of precipitation (PoP). The black curve is the conditional PDF given that precipitation is nonzero, the dashed vertical line on the left (right) is the 10th (90th) percentile of the total PDF, the black dot is the verifying observation, the blue dot is the ensemble mean forecasts, and the red diamond is the deterministic forecast of BMA.

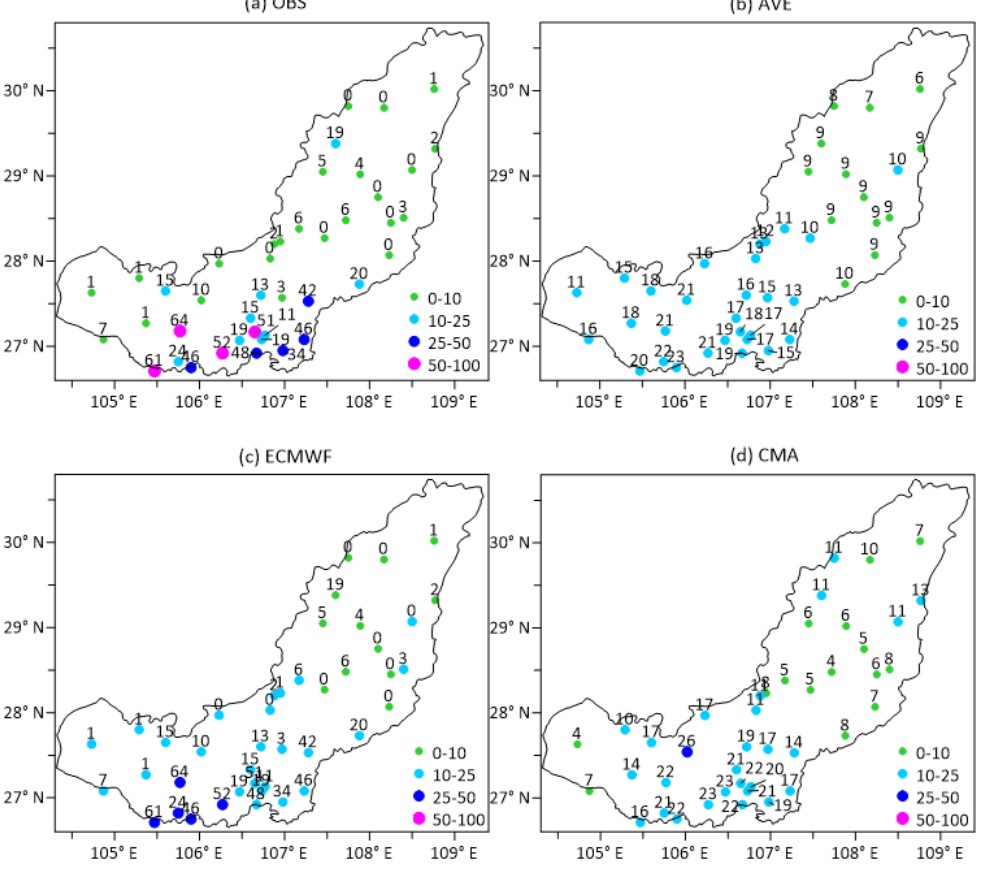

**Figure 7.** *Cont.*

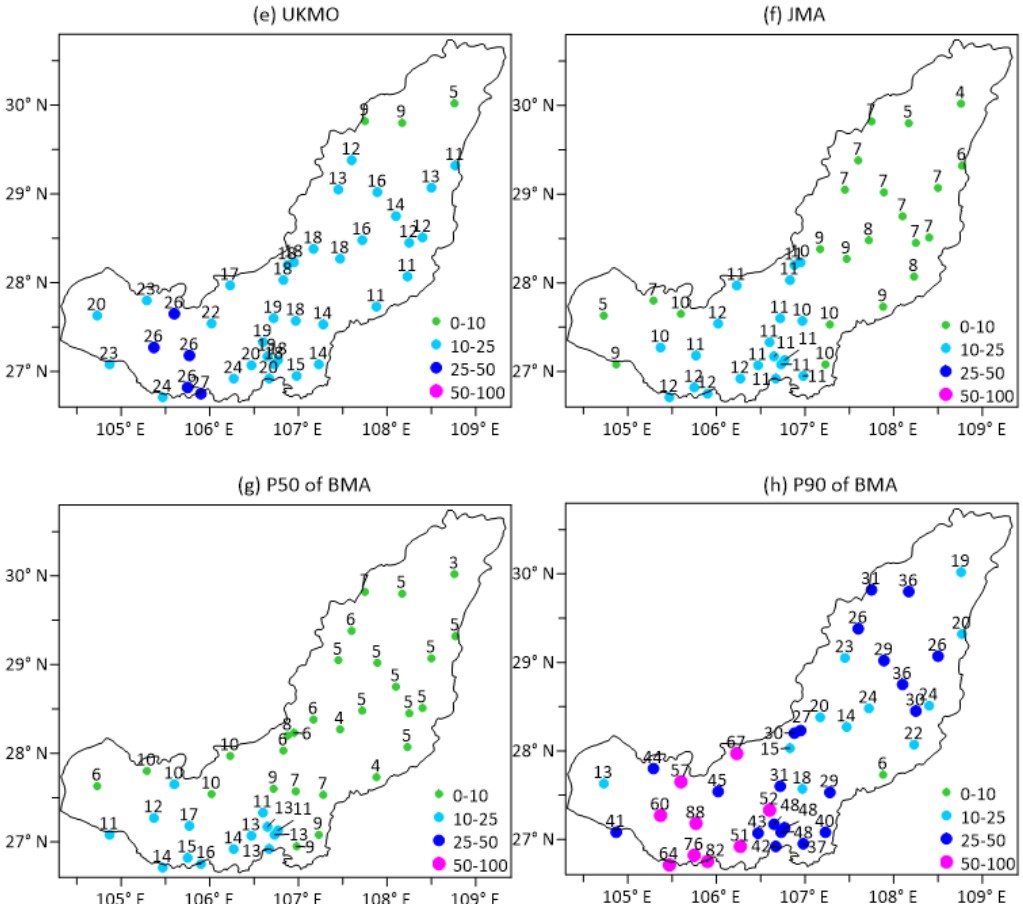

**Figure 7.** The forecast results and their corresponding observed precipitation distribution (**a**) ensemble average forecasting (**b**) single model (**c**–**f**), 50th percentile (**g**), and 90th percentile (**h**) of BMA forecasts. The numerals are the observed precipitation. The colored dots represent the levels of 24 h accumulated observed precipitation, including the light rain (0–9.9 mm, green dots), moderate rain (10–24.9 mm, lake blue dots), heavy rain (25–49.9 mm, blue dots) and rainstorm (50–100 mm, pink dots).

Based on the above analysis, it can be concluded that the PDF produced by BMA can control the range of the forecast uncertainty. Moreover, through an analysis of the full PDF, the quantitative forecasts can be provided using the percentile forecast data. For heavy precipitation forecast, especially precipitation over rainstorm level, it is recommended to refer to forecast results of the 75th–90th percentiles, which are more reasonable. Nevertheless, extreme events with low probability may occur and cannot be ignored. As for normal light precipitation, BMA deterministic forecast results are more reliable and can be used for reference.

### 4.4. Verifications over the Season

Here, two indicators including MAE and CRPS were used to evaluate the ensemble forecasting products of four centers and BMA in the Wujiang River Basin in summer 2018 for the verification period. In the results of MAE and CRPS, MAE of BMA reduced nearly 31% compared with that of the best single center JMA (Figure 8a), while CRPS of BMA improved by nearly 9% compared with ECMWF (Figure 8b).

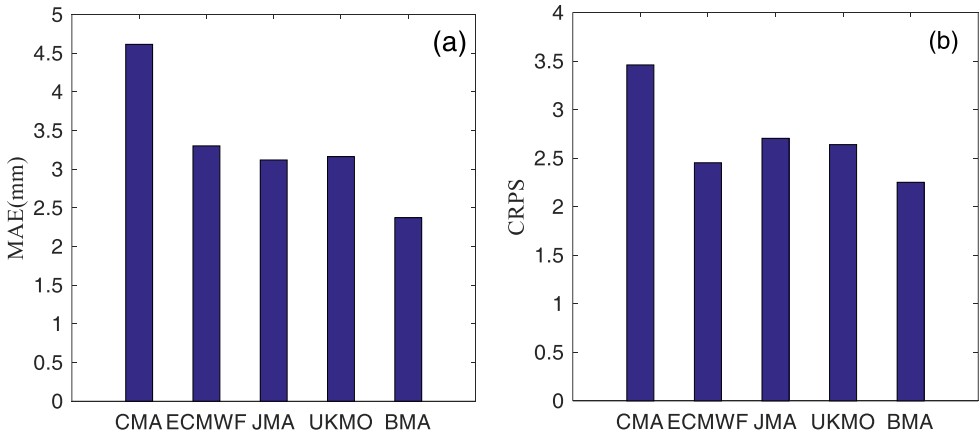

**Figure 8.** (**a**) MAE and (**b**) continuous ranked probability score (CRPS) from the four centers and BMA during 21 July to 31 August 2018.

In summary, the forecasting effects of BMA had a further improvement in predicting the certainty and probabilistic forecasting testing indicators compared with the centers. This method can also provide a certain technical support for business forecasting work.

## 5. Conclusions and Discussion

In this paper, based on the precipitation data of a TIGGE multi-center ensemble forecasting system, two multi-model ensemble probability forecasting methods of BMA and LR were selected for the comparative study. Meanwhile, the performances of two models for heavy rain events forecast over the mountainous area of Wujiang River Basin in the summer of 2018 were compared. The main conclusions are as follows.

1.  The BMA forecasting model was sensitive to the length of the training period and the forecast quality of the model around the forecast period. Additionally, for the forecast skill, a longer training period is not necessarily better. The training period of 2 years was the best for BMA, whereas the LR model required more statistical samples compared with the BMA method. The LR prediction effect was optimal when the length of the training period was 5 years.
2.  The multi-model ensemble prediction method was not always superior to RAW under any lead time or at any precipitation level. According to the BS, for precipitation events exceeding 10 mm with lead times of 1–7 days, the BMA forecasting technique outperformed the LR and the RAW with lead times of 2–7 days, while for heavy precipitation events exceeding 25 mm and 50 mm, the forecast skill of RAW was equivalent to that of BMA, whose improvement is little. This can be due to two reasons. First, the four-model multi-member ensemble average forecast used in this study was equivalent to the BMA ensemble of the four members. Compared with the multiple and comprehensive forecast results obtained by the four models with 131 members, the improvement was greatly limited. Second, the correcting effect of the BMA method itself has certain limitations. According to the verification results, for heavy and moderate rainfall events, forecasts of BMA and the LR were more reliable than those of the RAW, and the BMA model shows the best performance.
3.  The LR and the RAW could only provide the probability forecast at a certain threshold, while the BMA had the advantage of producing a highly concentrated full PDF curve and rendering the deterministic forecast. The PDF curve can control the uncertainty range of forecast results, and can also render quantitative forecasts through analyzing the percentile data. With respect to heavy precipitation forecasting, it is recommended to refer to forecasting results at the 75th–90th percentiles, which are more reasonable. Nevertheless, extreme weather events with low probability

forecasts cannot be ignored because they might also occur. As for the light precipitation event, the BMA forecast results at the 50th percentile were closer to the observation.

It is worth noting that during the training, NCEP data was used to replace CMA data and UKMO data and some errors may have occurred in parameter estimation during the training. This study only conducted a comparative forecasting study for the summer of 2018. Therefore, the sample data is limited and the study has certain limitations. In the future, more large-sample data can be obtained and evaluated. Determining how to properly apply the BMA forecasting model in the flood forecasting system is the next research work.

**Author Contributions:** Conceptualization, X.Z. and H.Q.; Methodology, H.Q.; Software, H.Q.; Validation, X.Z.; Formal analysis, H.Q. and X.Z.; Data curation, C.L. and Y.B.; Writing—original draft preparation, H.Q.; Writing—review and editing, T.P. and C.M; Supervision, X.Z. and T.P.; Project administration, T.P.; Funding acquisition, Y.B.

**Funding:** This research was funded by the National Key Research and Development Program of China (2018YFC1507200, 2018YFC1508002), Hubei Provincial Natural Science Foundation (2018CFB706), the Research and Operation Project of the Institute of Heavy Rain, CMA, Wuhan (IHRKYYW20190, IHRKYYW201911).

**Acknowledgments:** The authors would like to thank Peng Ting, Ji Luying and Meng Cuili for providing valuable suggestions. The authors are also grateful for support from the Open Foundation of Jiangsu Key Laboratory of Agricultural Meteorology, Nanjing University of Information Science & Technology (JKLAM1503), and the Research and Operation Project of the Institute of Heavy Rain (IHRKYYW201807).

**Conflicts of Interest:** The authors declare no conflict of interest.

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
