# Peer review of "Comparative Study on Probabilistic Forecasts of Heavy Rainfall in Mountainous Areas of the Wujiang River Basin in China Based on TIGGE Data"

_atmosphere, doi:10.3390/atmos10100608_

Round 1
Reviewer 1 Report
Please see the attachment for my review.

Reviewer 2 Report
See attached.

Round 2
Reviewer 2 Report
See attachment. The authors did little to address my first round of concerns in this revision.

Author Response
Responses are attached.

Round 3
Reviewer 2 Report
Below are my remaining comments on this paper: Response to response 1 - I still think their discussion on the model data selected (NCEP vs. UKMO, etc.) is too vague as there are numerous NCEP models and they are not providing the detailed information needed. I am assuming they are working with the GFS model product from NCEP, which is global and on a 0.5 degree grid. It could be a different product though, not sure. Response to their response 3 - Without optimizing the logistic regression model you're basically guaranteeing BMA will be the superior method just because of how the model was created. If the logistic regression results are to be included they need to justify their selection of the ensemble mean as the sole predictor for logistic regression. The issue may be context; if the logistic model is a control model to compare against BMA then it is easier to defend using the ensemble mean as the simplest logistic model. If however the authors are claiming the logistic model provides a quality forecast (which it won't necessarily do using just the ensemble mean), they need to optimize the logistic model before the paper can be published. Response to their response 4 - Just because the reliability diagram forecast frequencies match the observational frequencies does not guarantee the forecast is correct, only that it is reliable, which in proper context means it is unbiased. The diagram itself says nothing about verification of individual events, only the frequency of occurrence of different values in the observations and in the predictions. Also the Fig. 6 caption in the response never mentions the boxplot that was generated using bootstrapping. This should be added.Author Response
Please see the attachment.
